

**Moment tensor catalogue of microearthquakes in West Bohemia from**
**2008 to 2018**
Václav Vavryčuk[1], Petra Adamová[1], Jana Doubravová[1], Josef Horálek[1]
[1]Institute of Geophysics, Boční II/1401, 14100 Praha 4, Czech Republic
**Correspondence:** Václav Vavryčuk (vv@ig.cas.cz)
**Abstract**
We present a unique catalogue of full moment tensors (MTs) of microearthquakes that occurred in West
Bohemia, Czech Republic, in the period from 2008 to 2018. The catalogue is exceptional in several aspects:
(1) it represents an extraordinary extensive dataset of more than 5.000 MTs, (2) it covers a long period of
seismicity in the studied area, during which several prominent earthquake swarms took place, (3) the locations
and retrieved MTs of microearthquakes are of a high accuracy. Additionally, we provide three-component
records at the West Bohemia (WEBNET) seismic stations, the velocity model in the region, and the technical
specification of the stations. The dataset is ideal for being utilized by a large community of researchers for
various seismological purposes, e.g., for studies of (1) the migration of foci and the spatiotemporal evolution
of seismicity, (2) redistribution of stress during periods of intense seismicity, (3) the interaction of faults, (4)
the Coulomb stress along the faults and local stress anomalies connected to fault irregularities, (5) diffusivity
of fluids along the activated faults, or (6) the time-dependent seismic risk due to the migration of seismicity in
the region. In addition, the dataset is optimum for developing and testing new inversions for MTs and for
tectonic stress. Since most of the earthquakes are non-shear, the dataset can contribute to studies of non-double-
couple components of MTs and their relation to shear-tensile fracturing and/or seismic anisotropy in the focal
zone.


**1 Introduction**
The seismic moment tensor (MT) describes equivalent body forces acting at an earthquake source (Knopoff
and Randall, 1970). It is a basic quantity evaluated for earthquakes that informs us about their moment
magnitude, focal mechanism and type of faulting. It is formed by double-couple (DC), isotropic (ISO) and
compensated linear vector dipole (CLVD) components (Jost and Hermann, 1998; Vavryčuk, 2015). The DC
component is produced by shear faulting in isotropic media; the ISO and CLVD components reflect
complexities in the earthquake source, e.g., irregularly shaped faults, seismic anisotropy, shear-tensile faulting
induced by fluid injection in volcanic or geothermal areas, or the presence of a material interface in the focal



zone (Frohlich, 1994; Julian et al. 1998; Miller et al. 1998; Šílený and Milev, 2008; Vavryčuk  2005, 2006,
2011a, 2013, 2015; Vavryčuk and Hrubcová 2017).

Since earthquakes do not occur separately but in sequences, it is necessary to compile high-quality MT
catalogues for understanding origins of seismicity, tectonic stress regime and seismic energy release of any
region under study. In this way, we can identify prominent periods of seismicity, trace faults and fault
segments, monitor migration of earthquake foci, analyse interactions of nearby or intersecting faults, and map
the fluid flow along the fault systems in the focal zone (Vavryčuk et al., 2021). Hence, MT catalogues are
fundamental sources of information for all detailed studies of seismicity on the local, regional or global scale.

In this paper, we present recordings, locations and high-quality moment tensors of 5182 microearthquakes that
occurred in the West Bohemia geothermal region, Czech Republic in the period from 2008 to 2018. The
microearthquakes were monitored by the West Bohemia local seismic network WEBNET (Horálek et al.,
2000; Fischer et al., 2010). Their locations were calculated by the double-difference location method and the
moment tensors were determined using the moment tensor inversion of P waves based on the principal
component analysis. Because of its extent and quality, the presented dataset is unique and represents an
extraordinary dataset, which might find exciting applications in numerous future studies.

**2 West-Bohemia seismoactive region**
The region of West Bohemia is located in the western part of the Bohemian Massif, where three major tectonic
units are merged: the Saxothuringian, the Teplá-Barrandian and the Moldanubian. The region is
geodynamically active exposed to the Tertiary and Quaternary volcanism associated with $CO_2$ emanations,
dry and wet mofettes, and numerous mineral springs (Kämpf et al., 2013; Hrubcová et al., 2017; Bräuer et al.,
2018). Two major fault systems are identified in the area: the Mariánské Lázně fault striking in the NW-SE
direction and the Ore-Mountain fault striking in the WSW-ENE direction (Figure 1b). The recently most active
fault is, however, a left-lateral strike-slip fault in the N-S direction, situated at the eastern boundary of the Cheb
Basin filled by up to 300 m thick Tertiary and Quaternary sediments. The seismically active faults were
identified at depth by clustering of hypocentres and by focal mechanisms (Vavryčuk et al., 2013), but they also
have some geological evidence on the surface (Bankwitz et al., 2003).

The seismic energy in the West Bohemia region is typically released in the form of earthquake swarms. The
occurrence of the earthquake swarms has been well documented in the region since the beginning of the 19th
century. A significant increase of the earthquake activity was observed at the turn of the 19th and 20th century,
when several larger swarms were observed. There were earthquake swarms in 1897, 1900, 1903 and 1908.
During the last 40 years, the seismicity occurs in the area of 40 x 50 square kilometres, but the most intense
seismicity is focused in the Nový Kostel zone with size of 3 x 12 square kilometres (Fischer et al., 2014;
Čermáková and Horálek, 2015). Foci of microearthquakes in this zone are clustered along a fault striking in
the roughly N-S direction (Figure 1a) with depths ranging from 6 to 11 km. The duration of the earthquake
swarms varies; it lasts from several days for micro-swarms up to 2-3 months for the most prominent swarms.
The swarms may consist of several thousands of microearthquakes. The local magnitudes $M_L$ of the
microearthquakes rarely exceed a value of 4.0. The strongest instrumentally recorded swarm activity occurred
in 1985/86 with two main shocks having magnitudes of $M_L$ 4.6 and 4.2.

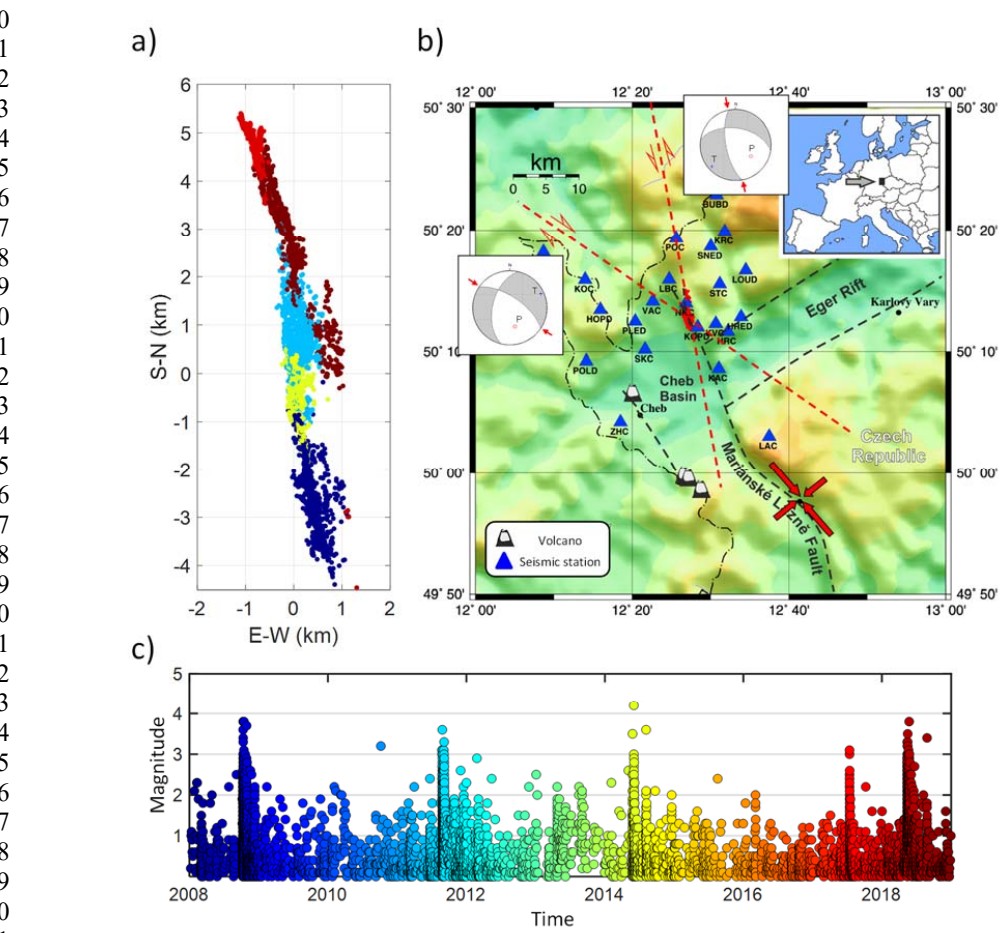

**Figure 1.** (a) The map view of earthquake foci in the period from 2008 to 2018, (b) topographic map with tectonic faults (black dashed lines) and positions of stations (blue triangles), and (c) the magnitude-time plot with the colour-coded time. The red dots in (b) show the earthquake foci. The red full arrows mark the orientation of the maximum and minimum principal stress axes. The dashed-dotted line marks the boundary between the Czech Republic and Germany. The position of West Bohemia in Europe is indicated in the inset. The focal mechanisms typical for the area are also indicated.



## 3 Monitoring system

The seismic activity in the region is monitored by the local seismic network WEBNET (Figure 1b, Table 1). The network was operating since 1994 and the number of stations gradually increased (Horálek et al., 2000; Fischer et al., 2010). After its major upgrade in 2008, the WEBNET network consists of 23 seismic stations within the epicentral distance of 25 km. The stations cover the area uniformly with no azimuthal gaps. The three component ground-velocity records are sampled at 250 Hz and the frequency response is flat at least between 1 and 80 Hz. Until September 2014, all data were processed based on triggered records. Since the beginning of September 2014, the recordings are processed by using automatic pre-processing of continuous recordings. Another major upgrade of the network was realized in 2015. Originally, the stations were equipped by the Le-3DLite and SM3 seismometers; some of them were lately upgraded using the Guralp CMG-3ESP seismometers. The station with the nearest epicentral distance (station NKC) is additionally equipped with the broadband STS-2 seismometer. For a detailed technical specification of the WEBNET seismic stations, see Table 1.

**Table 1.** Location and instrumentation of the WEBNET seismic stations

| Code | Site name | Latitude (°N) | Longitude (°E) | $h$ (m) | Sensor before 2015 | Digitizer before 2015 | Sensor after 2015 | Digitizer after 2015 | Note |
|------|-----------|---------------|----------------|---------|--------------------|-----------------------|-------------------|----------------------|------|
| BUBD | Bublava | 50.38174 | 12.51362 | 746 | LE-3DLite | Gaia | LE-3DLite | Gaia | |
| HOPD | Horní Paseky | 50.22378 | 12.26547 | 731 | LE-3DLite | Gaia | LE-3DLite | Gaia | |
| HRC | Hrádek | 50.19348 | 12.53660 | 596 | LE-3DLite | Gaia | LE-3DLite | Gaia | Out of order from 2015 |
| HRED | Hřebeny | 50.21425 | 12.56491 | 589 | LE-3DLite | Gaia | LE-3DLite | Gaia | |
| HUC | Komorní Hůrka | 50.09997 | 12.33612 | 480 | - | - | CMG-3ESPC | Taurus | Installed in 2016 |
| KAC | Kaceřov | 50.14361 | 12.51708 | 548 | SM-3 | Janus-Trident | SM-3 | Janus-Trident | |
| KOC | Kopaniny | 50.26417 | 12.23288 | 621 | SM-3 | 5800 PCM | CMG-3ESPC | Centaur | |
| KOPD | Kopanina | 50.20319 | 12.47473 | 536 | LE-3DLite | Gaia | LE-3DLite | Gaia | |
| KRC | Kraslice | 50.33069 | 12.52950 | 806 | SM-3 | Janus-Trident | CMG-3ESPC | Centaur | |
| KVC | Květná | 50.20496 | 12.51134 | 666 | SM-3 | 5800 PCM | CMG-3ESPC | Centaur | |

| LAC | Lazy | 50.04967 | 12.62396 | 884 | SM-3 | 5800 PCM | CMG-3ESPC | Centaur | |
| LBC | Luby | 50.26461 | 12.41123 | 684 | SM-3 | Janus-Trident | CMG-3ESPC | Centaur | |
| LOUD | Loučná | 50.27753 | 12.57449 | 692 | LE-3DLite | Gaia | LE-3DLite | Gaia | |
| NKC | Nový Kostel | 50.23234 | 12.44706 | 610 | SM-3 CMG-40T | 5800 PCM Janus-Trident | CMG-3ESPC STS-2 | Centaur | |
| PLED | Plesná | 50.20890 | 12.33767 | 556 | LE-3DLite | Gaia | LE-3DLite | Gaia | |
| POC | Počátky | 50.31997 | 12.42662 | 841 | SM-3 | Janus-Trident | CMG-3ESPC | Centaur | |
| POLD | Polná | 50.15603 | 12.23497 | 556 | LE-3DLite | Gaia | LE-3DLite | Gaia | |
| SKC | Skalná | 50.16911 | 12.36050 | 501 | SM-3 | Janus-Trident | CMG-3ESPC | Centaur | |
| SNED | Sněžná | 50.31088 | 12.50131 | 756 | LE-3DLite | Gaia | LE-3DLite | Gaia | |
| STC | Studenec | 50.25794 | 12.51849 | 712 | SM-3 | Janus-Trident | CMG-3ESPC | Centaur | |
| TRC | Trojmezí | 50.30344 | 12.14466 | 612 | LE-3DLite | Gaia | CMG-3ESPC | Centaur | |
| VAC | Vackov | 50.23450 | 12.37634 | 581 | SM-3 | Janus-Trident | CMG-3ESPC | Centaur | |
| ZHC | Zelená Hora | 50.06984 | 12.30810 | 677 | CMG-40T | Janus-Trident | CMG-3ESPC | Centaur | |
| MAC | Chlum sv. Maří | 50.14429 | 12.53516 | 609 | - | - | CMG-3ESPC | Centaur | Installed in 2017 |

Quantity *h* means the altitude of the stations. Recording systems: Taurus – Nanometrics digitizer; Janus-Trident
– Nanometrics communications controller-digitizer; Centaur – Nanometrics digitizer; Gaia – Vistec digitizer;
5800 PCM – Lennartz digitizing system. Seismometers: SM-3 – SP sensor; LE-3DLite – Lennartz SP sensor;
CMG-40T – Guralp BB sensor; CMG-3ESPC – Guralp BB sensor.

## 4 Seismicity in 2008-2018

The West Bohemia region is characterized by a continuous background seismicity scattered over the whole
region interrupted by earthquake swarm sequences located mostly in the Nový Kostel focal zone. The most
intense periods of seismicity in 2008, 2011, 2014, 2017 and 2018 (Figures 1c and 2). All these sequences are
typical earthquake swarms except for the seismic activity in 2014, which was exceptional. This sequence
resembled a mainshock-aftershock sequence rather than the earthquake swarm (Hainzl et al., 2016; Jakoubková
et al., 2018; Vavryčuk and Adamová, 2018) being formed by three pronounced activity periods. The strongest
events in these periods reached magnitude significantly larger than the other events (Figure 2c). The seismic
sequences differ in the earthquake productivity, in the duration, and in the number of periods of the intense
seismicity (Figure 2). The strongest event in the period from 2008 to 2018 reached magnitude $M_L$ of 4.2 and
it occurred in 2014.

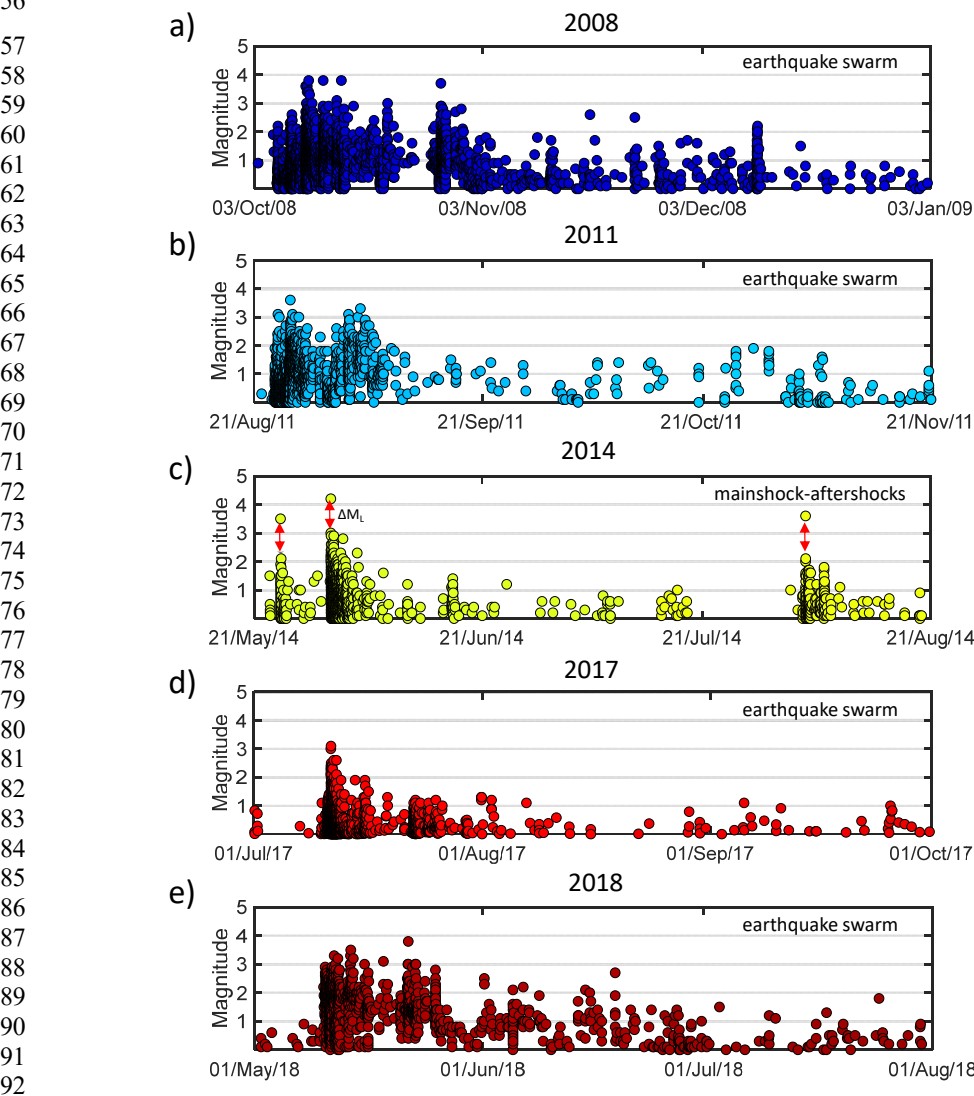

**Figure 2.** Magnitude-time plots of the major seismic sequences in the period from 2008 to 2018. According to
the Bath law (Bath, 1965), the 2014 activity resembles rather a mainshock-aftershock sequence, because the
difference in magnitudes $\Delta M_L$ between two strongest events in individual seismicity phases exceeds 1. In other
seismic sequences, the magnitude gaps between two strongest events are not so prominent.





## 5 Magnitudes and foci locations

The local magnitude of earthquakes is computed from the velocity records according to the formula of Horálek et al. (2000). The locations are computed in two steps. First, initial locations were calculated by the NonLinLoc code (Lomax et al., 2009) in a layered velocity model (see Table 2) developed by Málek et al. (2005). For the locations, manual picks of the P and S arrivals were used. Second, we applied the double-difference location algorithm developed by Waldhauser and Ellsworth (2000) to differential times calculated from manual picks. The relative precision of hypocentres was less than ±20 m within the cluster (Bouchaala et al., 2013). The absolute location of the cluster was determined with the accuracy of about ±100 m in the horizontal plane and ±350 m in depth (see Bouchaala et al., 2013).

The locations of foci point to complex geometry of the fault system in the focal area (Figure 3). The seismicity migrated from south to north in time and the individual seismic sequences occurred along different subfaults (Fischer et al., 2010; Bouchaala et al., 2013; Vavryčuk et al., 2013; Jakoubková et al., 2017). For example, the 2008, 2011 and 2017 swarms activated three similarly oriented subfaults separated with gaps and offsets between them. The barrier between the fault segments activated in 2008 and 2011 was broken in 2014 (Hainzl et al., 2016; Vavryčuk and Adamová, 2018), and the gap between the fault segments activated in 2011 and 2017 was broken during the 2018 swarm (Bachura et al., 2021; Vavryčuk et al., 2021). The overall direction of the whole fault system is defined by strike of 170° and dip of 75°. However, some fault segments may deviate from this overall direction significantly. For example, small echelon faults located at the deepest part of the fault system have strike of 305° and dip of 65° (see Figure 3c, blue dots at the depth range of 10.5-11 km).

**Table 2.** The layered velocity model

| Depth (km) | 0.0 | 0.2 | 0.5 | 1.0 | 2.0 | 4.0 | 6.0 | 10.0 | 20.0 | 32.0 |
|---|---|---|---|---|---|---|---|---|---|---|
| $v_P$ (km/s) | 4.30 | 5.06 | 5.33 | 5.60 | 5.87 | 6.09 | 6.35 | 6.74 | 7.05 | 7.25 |
| $Q_P$ | 30 | 40 | 50 | 60 | 80 | 100 | 150 | 200 | 300 | 400 |

Ratio $v_P/v_S$ is 1.70 and ratio $Q_P/Q_S$ is 2.

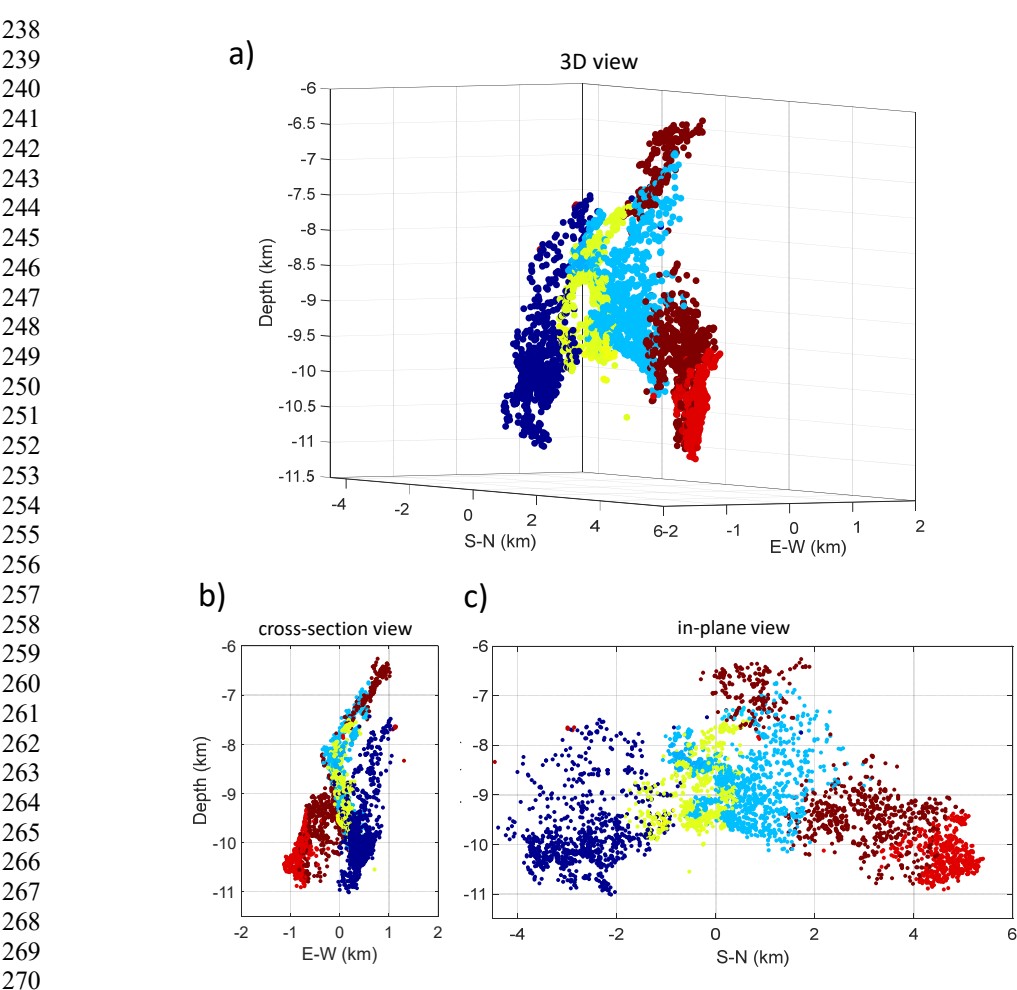

**Figure 3.** The earthquakes foci with local magnitude $M_L \geq 0.5$ in (a) 3D view, (b) cross-section vertical view, and (c) in-plane vertical view. The foci are colour coded according to time: dark blue – 2008, light blue – 2011, yellow – 2014, red – 2017, and brown – 2008.

## 6 Moment tensors

### 6.1 MT inversion of microearthquakes

The MT inversion requires accurate locations of earthquakes, an accurate crustal velocity model, dense coverage of stations on the focal sphere and low seismic noise (Šílený, 2009; Ford et al., 2010; Stierle, Bohnhoff, et al., 2014; Stierle, Vavryčuk, et al., 2014). We can invert amplitudes of seismic phases, amplitude ratios or full waveforms (Dreger and Woods, 2002; Cesca et al., 2006; Sokos and Zahradník, 2008; Cesca and Dahm, 2008; Vavryčuk et al., 2008; Zahradník at al., 2008; Fojtíková et al., 2010; Kwiatek et al., 2016;



Jechumtálová and Šílený, 2005; Vavryčuk and Kühn, 2012; Yu et al., 2018, 2019). Each inversion is applicable
to earthquake of different magnitudes, wave frequencies and epicentral distances of stations. MTs of moderate
or large earthquakes are usually calculated from full waveforms recorded at regional or global seismic
networks. By contrast, MTs of small earthquakes and microearthquakes are commonly calculated from
amplitudes of P and/or S waves picked in short-period seismograms recorded at local networks.
The inversion for MTs of microearthquakes is challenging for several reasons: (1) the waveforms are complex
due to high frequencies and noise, and (2) the datasets are extensive with thousands of events, which require a
semi- or fully-automated processing. Here, the MT inversion developed by Vavryčuk et al. (2017) is applied.
The inversion is based on the principal component analysis (PCA), which transforms correlated waveforms
into a set of the called principal components (see Figure 4). The first component has the highest variance and
reproduces a so-called 'common wavelet', i.e., a wavelet with the highest similarity with all analysed traces.
This common wavelet physically represents a signal radiated by the earthquakes source, which can be distorted
during its propagation from the source to the receiver by inhomogeneities in the geological structure, site
effects or seismic noise.
Subsequently, the common wavelet is correlated with individual recorded traces and the effective P-wave
amplitudes are calculated as the amplification factors applied to the common wavelet, in order to optimally
reproduce the recorded traces. The obtained amplitudes are inverted for the MTs using the generalized linear
inversion (Lay and Wallace, 1995). The Green's function amplitudes are computed by the ray method
(Červený, 2001) and incorporated the effects of the Earth's surface. An inhomogeneous medium with a vertical
gradient obtained by smoothing the layered model of Málek et al. (2005) was applied for computing the rays
by the ray-tracing algorithm. The inversion is robust, fast and insensitive to noise in data.
**6.2 Individual steps of the MT inversion**
The MT inversion consists of data pre-processing, alignment of traces, computation of the effective amplitudes
using the PCA method and the MT inversion. The individual steps of the inversion are as follows (see Figure
5):
1. Data pre-processing, which comprises: (a) an oversampling of records in order to perform an accurate
alignment of waveforms, (b) band-pass filtering to enhance the signal-to-noise ratio, and (c) a rough
alignment of waveforms using manual picks, if available, or using an automatic picking algorithm called
the Suspension Bridge Picking (SBPx), see FeedMeImATroll (2021).

2. Two-step accurate alignment of waveforms, which comprises: (a) an alignment of waveforms using the
cross-correlation with the waveform of the highest signal-to-noise ratio, (b) calculation of the first principal
component from the aligned waveforms, (c) another alignment of waveforms using the cross-correlation
with the computed first principal component, and (d) calculation of the refined first principal component
from the aligned waveforms.

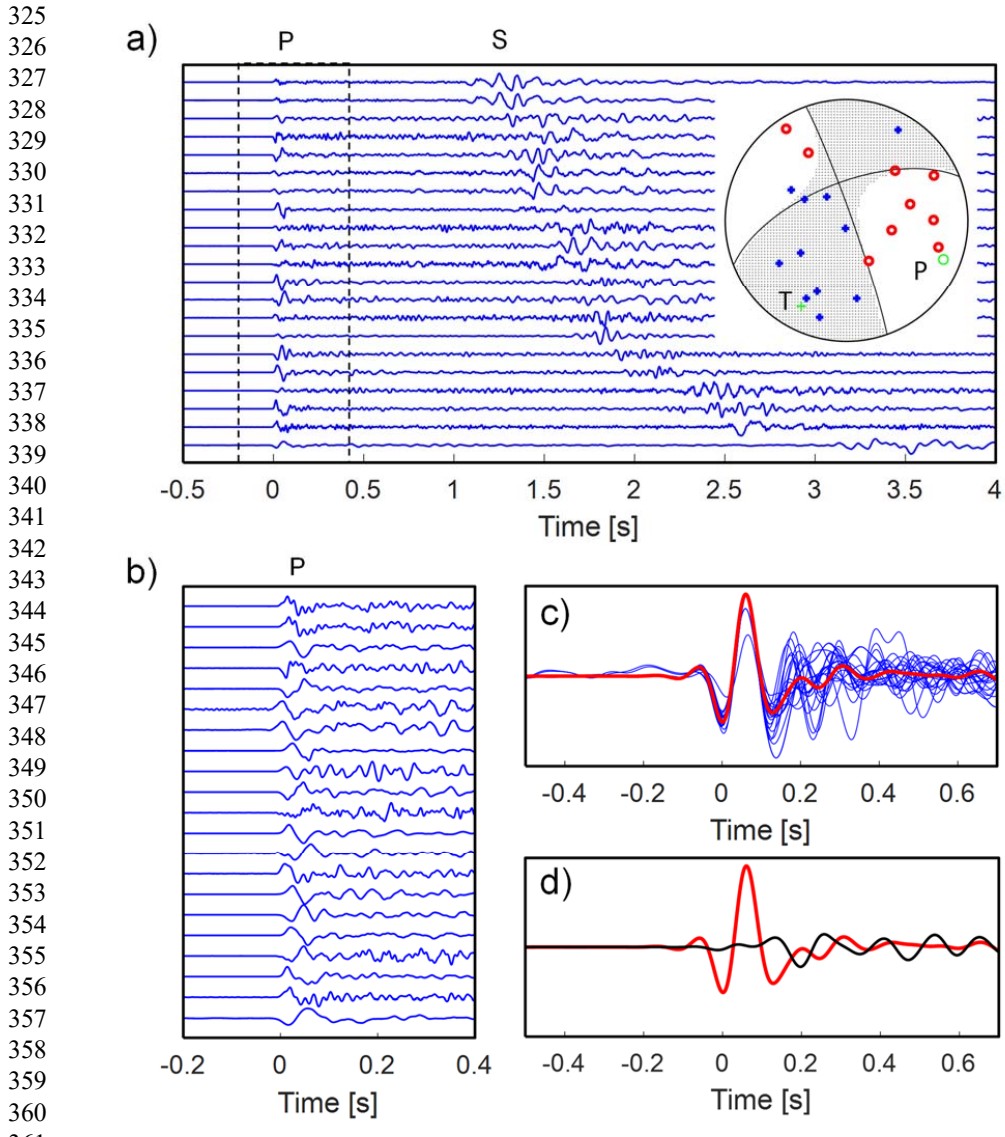

**Figure 4.** Example of the MT inversion of the microearthquake on 24 May 2014 at 16:14:30 with ML 2.1. (a) Whole velocity records; (b) window with aligned P waves; (c) the common wavelet (red line) together with the P-wave traces at individual stations (blue lines); (d) the common wavelet represented by the first principal component (red line) and noise in waveforms represented by the second principal component (black line). The polarities of the P-wave in panel (c) are switched to be consistent with the polarity of the common wavelet. The inset in plot (a) shows the focal mechanism and positions of stations on the focal sphere (red circles mark negative polarities, and blue plus signs mark the positive polarities).

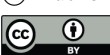



3. Calculation of the PCA amplitudes and weights in the MT inversion, which comprises: (a) calculation
of the PCA coefficients of the first principal component, which serve as the effective amplitudes used in
the MT inversion, (b) calculation of the correlation coefficients between individual traces and the first
principal component, which serve as the weights in the linear MT inversion scheme (in this way, a station
with a waveform significantly different from the common wavelet suppressed in the inversion),
4. Repeated MT inversion for several alternative band-pass filters and time windows, in order to adapt the
inversion to earthquakes with a varying frequency content. The inversion is firstly run with the whole set
of stations, and secondly with eliminating two stations producing the largest misfits in the inversion.
In this way, we obtain a set of candidate MTs. The optimum MT is that with the minimum root-mean-squares
(RMS) of differences between the synthetic amplitudes $A^{synth}$ and the observed amplitudes $A^{obs}$
$$\text{RMS} = \frac{\sqrt{\sum_{i=1}^{N}\left(A_i^{synth}-A_i^{obs}\right)^2}}{\sqrt{\sum_{i=1}^{N}\left(A_i^{synth}\right)^2}},\qquad(1)$$

where $N$ is the number of stations. The optimum MT is normalized and expressed in a relative scale, because
it is computed from wave amplitudes but not from full displacement records. The scalar moment is obtained
by integrating the common (displacement) wavelet. The optimum MTs were further decomposed into the DC,
ISO and CLVD components according to Equations (6-10) of Vavryčuk (2015).
**Figure 5.** Flowchart of the PCA moment tensor inversion.

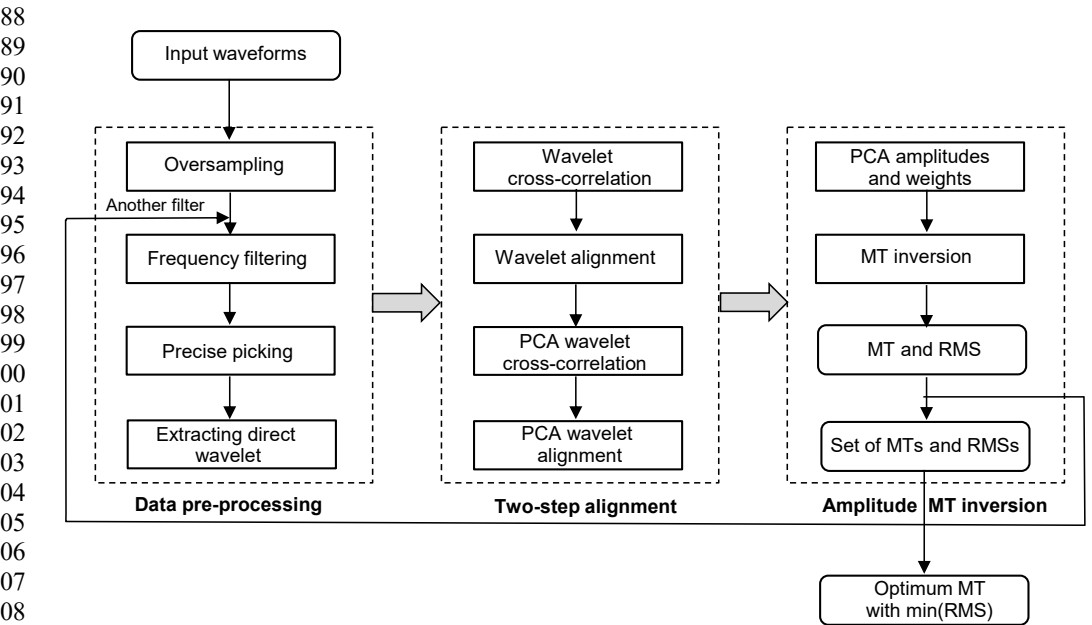

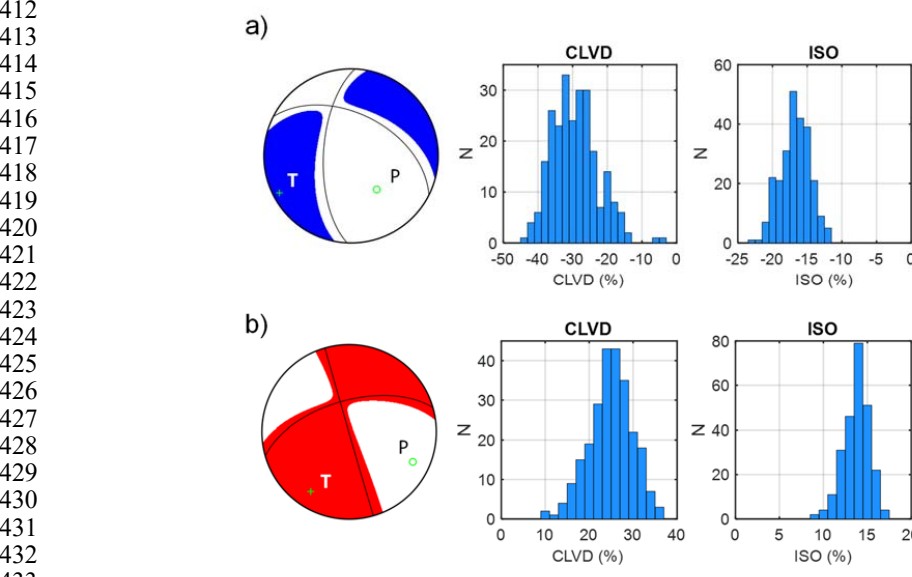

**Figure 6.** Examples of focal mechanisms and histograms of the CLVD and ISO errors. (a) Microerthquake on 1 September, 2011 at 12:54:05.7 with ML = 0.6, and (b) microearthquake on 11 May, 2018 at 06:26:09.0 with ML = 2.3.

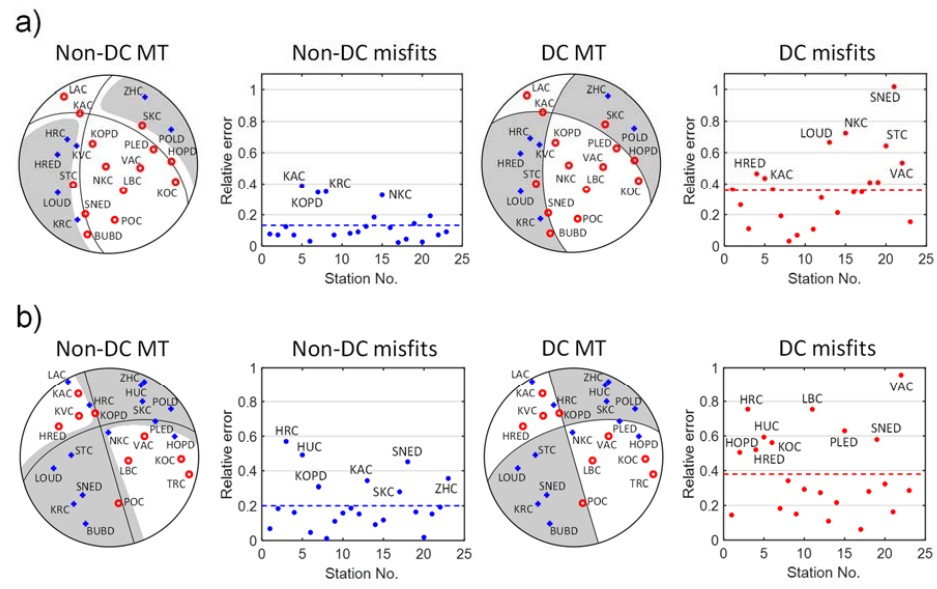

**Figure 7.** Inversion for the full MT solution ('Non-DC MT' and 'Non-DC misfits') and for the DC solution ('DC MT' and 'DC misfits') for microearthquakes in Figure 6. The mean amplitude misfits for the full MT and DC solutions are shown by blue and red dashed lines, respectively.

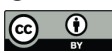



In order to estimate errors of the MTs, the inversion is performed for each MT repeatedly 100 times using
amplitudes distorted by noise characterized by a flat probability distribution. The level of noise ranges from -
25% to 25 % of the inverted amplitude at each trace. The scatter of the solutions served for estimating: (1) the
mean errors in the P/T axes directions calculated as the mean of deviations between the directions of the P/T
axes of noise-free solution and the noisy solutions, (2) the mean errors in the percentages of the DC, ISO and
CLVD components calculated as the standard deviations of the DC, ISO and CLVD values of noisy MT
solutions.

Figure 6 exemplifies the MT inversion for two micro-earthquakes, which display significant non-DC
components. The histograms of the CLVD and ISO errors indicate that the ISO component is always better
constrained than the CLVD component. Nevertheless, despite the numerical errors produced by the inversion,
the histograms prove that both the events contain also true non-DC components. This is also confirmed by a
comparison of fits for the full MTs and for the DC solutions for the events shown in Figure 7. The figure
indicates that the misfits for the full MT solutions are almost twice lower than those for the DC solutions. This
proves that at least some part of the non-DC components retrieved by the MT inversion should be of physical
origin.


**7 Basic characteristics of the MT catalogue**
Firstly, we processed all events with the local magnitude larger than 0.5. After that, we checked manually the
quality of input data and the retrieved MT and we excluded earthquakes: (1) recorded at a low number of
stations ($N < 14$), (2) with extremely low signal-to-noise ratio, (3) produced unstable moment tensors with
anomalously high RMS (RMS > 1). In this way, we obtained a dataset of 5182 earthquakes listed in the
catalogue. Table 3 summarizes the numbers of events in individual years. The magnitude-frequency
distribution of the analysed events is shown in Figure 8.
**Table 3.** Number of reported events for each year.

| Year | 2008 | 2009 | 2010 | 2011 | 2012 | 2013 |
|---|---|---|---|---|---|---|
| Number of events | 991 | 40 | 29 | 1225 | 69 | 201 |
| Year | 2014 | 2015 | 2016 | 2017 | 2018 | 2008-2018 |
| Number of events | 841 | 40 | 33 | 583 | 1130 | 5182 |




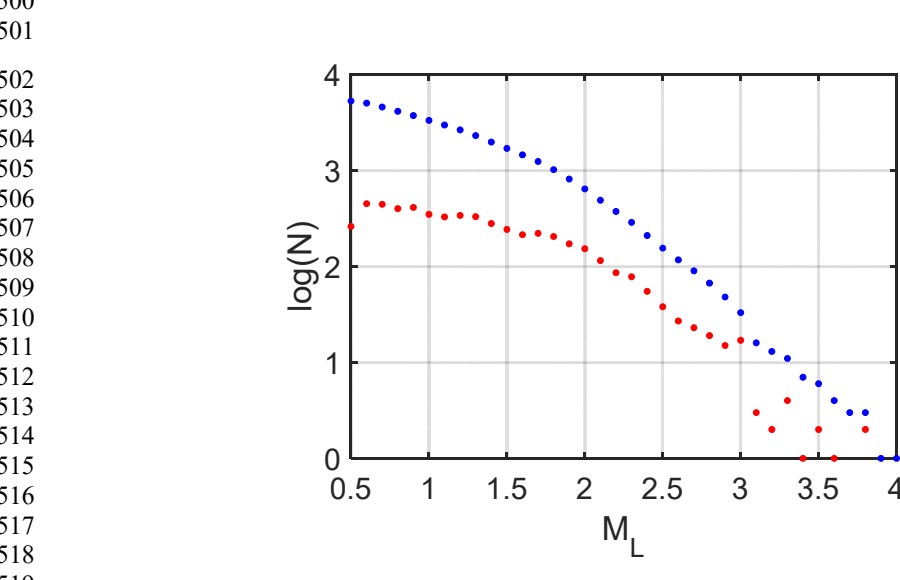

**Figure 8.** Cumulative (blue) and non-cumulative (red) magnitude-frequency distribution of the analysed earthquakes.

The earthquakes inverted for MTs were recorded mostly by 20 or more stations (Figure 9, middle column). The RMS varied during the whole period and ranged mostly from 0 to 0.5 (Figure 9, right column). The MTs with RMS higher than 0.5 were considered as unreliable. The variation of the RMS in time is probably produced by varying station coverage due to the foci migration. The P/T axes form compact and non-overlapping clusters for all seismic sequences in the studied time period (Figure 9). The position of clusters slightly differs in individual years and indicates some stress variation in the focal zone. Directions of the P/T axes are well resolved with the mean standard deviation less than 2° (Figure 10, two left columns). The errors of the ISO and CLVD components are mostly about 1.5-2% and 5-6%, respectively (Figure 10, two right columns). Comparing these errors for individual activity periods, we see that the errors tend to slightly decrease with time. This might be due to a continuously increasing quality of the WEBNET network. The histograms of the standard deviations of the P/T axes and the ISO and CLVD errors for the whole period from 2008-2018 are shown in Figure 11.

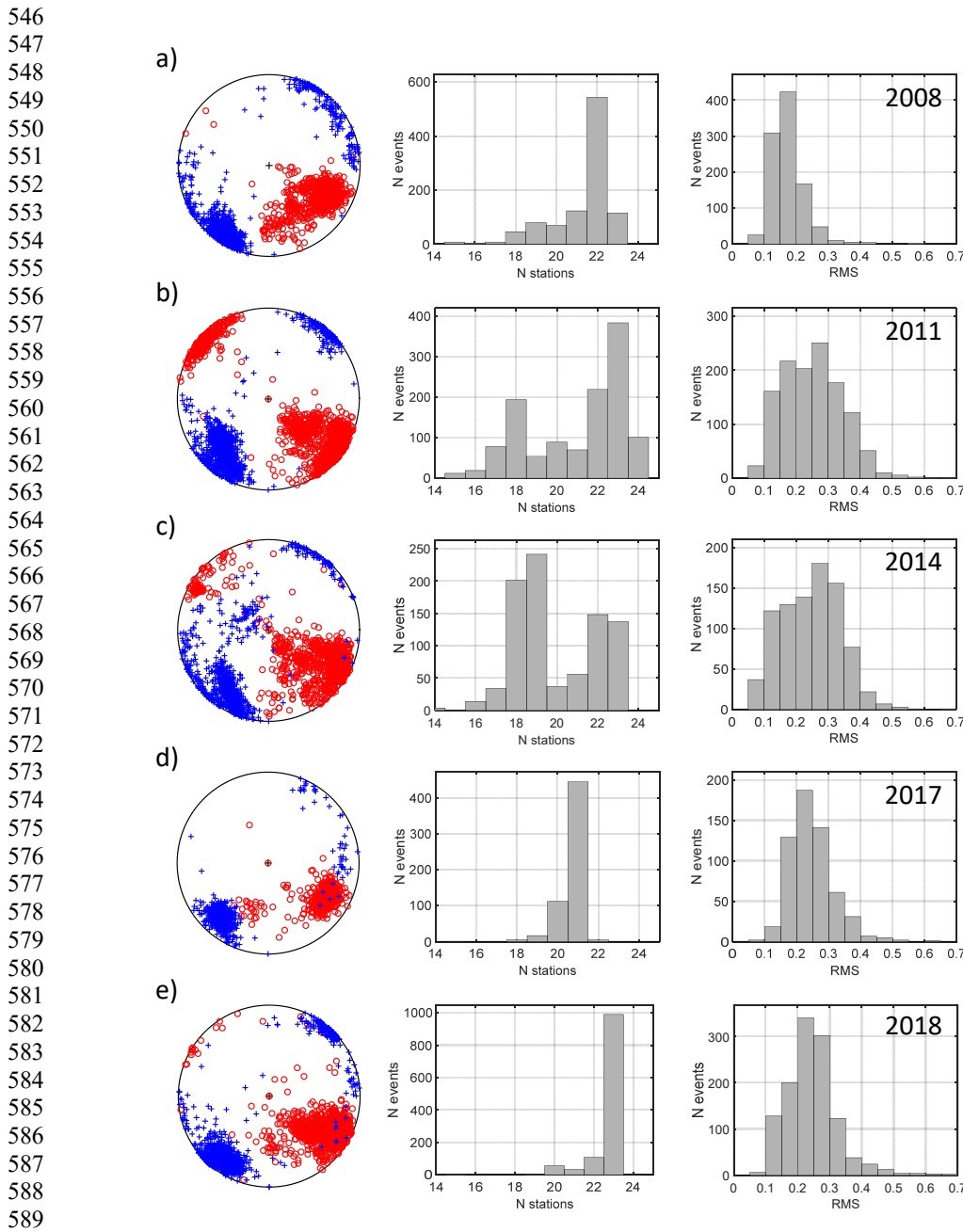

**Figure 9.** The P/T axes (left-hand plots), histograms of the RMS of the number of stations used in the MT inversion (middle plots), and histograms of the RMS of the retrieved MTs (right-hand plots) for seismic activities in 2008 (a), 2011(b), 2014 (c), 2017 (d) and 2018 (e). *N* denotes the number of stations, which recorded the individual earthquakes.

**Figure 10.** Histograms of mean deviations of the P/T axes and histograms of the ISO and CLVD standard errors for MTs of earthquakes from individual prominent seismic activities: in 2008 (a), 2011(b), 2014 (c), 2017 (d) and 2018 (e). The mean P/T deviations and the ISO and CLVD standard errors were calculated for each event from 100 MTs inverted using randomly generated noisy data.

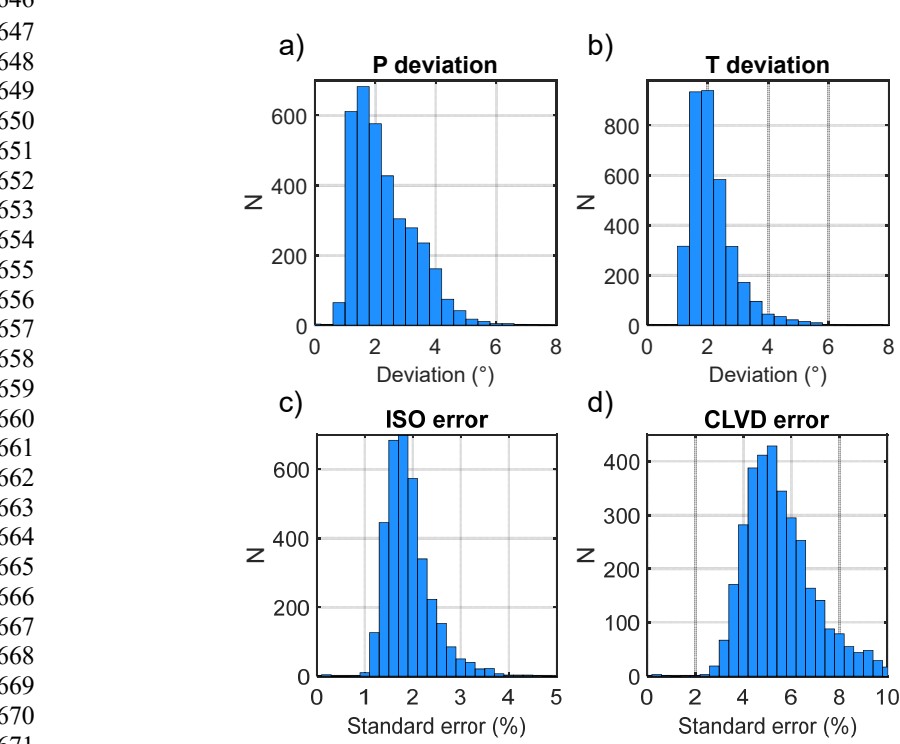

**Figure 11.** Histograms of mean deviations of the P/T axes (a-b) and histograms of the standard ISO and CLVD errors (c-d) for the 5182 reported MTs. The mean P/T deviations and the ISO and CLVD standard errors were calculated for each event from 100 MTs inverted using randomly generated noisy data.

## 8 Description of the dataset

The dataset consists of the following directories:

- Waveforms – this directory is further structured into subdirectories according to individual years and earthquakes. Three-component velocity records are stored in ASCII files with four columns (time + 3 components) individually for each station and each earthquake.

- Model – this directory contains the ASCII file 'model.crust', which defines the layered velocity model for the West Bohemia region (depth, P-wave velocity, $v_P/v_S$ ratio, $Q_P$ and $Q_P/Q_S$ ratio).

- Stations – this directory contains the ASCII file 'stations_Webnet.dat' with coordinates of stations (site, name of the station, latitude, longitude, elevation).

- Moments – this directory contains the ASCII file 'catalogue_2008-2018.dat' with double difference locations, magnitudes, moment tensors and their errors, RMS and the numbers of inverted stations.

- Figures – this directory is further structures into subdirectories according to individual years. Four figures are provided for each earthquake (see Figure 12): complete waveforms of vertical components,





a detail of the P-waveforms, the focal mechanism with positions of stations, and the RMS at individual
stations.
File 'catalogue_2008-2018.dat' lists the following quantities for each earthquake:
• Event identification (composed form year and the sequential number of the event in the year)
• Double-difference locations
• Origin time (year, day, hour, minute, second)
• Latitude (°N)
• Longitude (°E)
• Depth (km)
• Local magnitude $M_L$ (calculated according to Horálek et al., 2000)
• $N$ – number of stations used in the MT inversion
• Frequencies $f_1$ a $f_2$ (in Hz) – optimum parameters of the Butterworth band-pass filter
• RMS – for its definition, see Equation (1)
• Moment magnitude $M_W$
• Components of the normalized moment tensor: $M_{11}$, $M_{12}$, $M_{13}$, $M_{22}$, $M_{23}$, $M_{33}$ ($x_1$ – North, $x_2$ – East, $x_3$
– down). The moment tensor is normalized using the Euclidean norm (see Equation 17 of Vavryčuk,
2015)
• Strike1, dip1, rake1, strike2, dip2, rake2 (in °)
• DC, CLVD, ISO (in %, calculated according to Equations 6-10 of Vavryčuk, 2015)
• Errors of DC, CLVD, ISO (in %, for the definition of errors, see the text)
• Deviations of the P/T axes (in °, for the definition of errors, see the text)

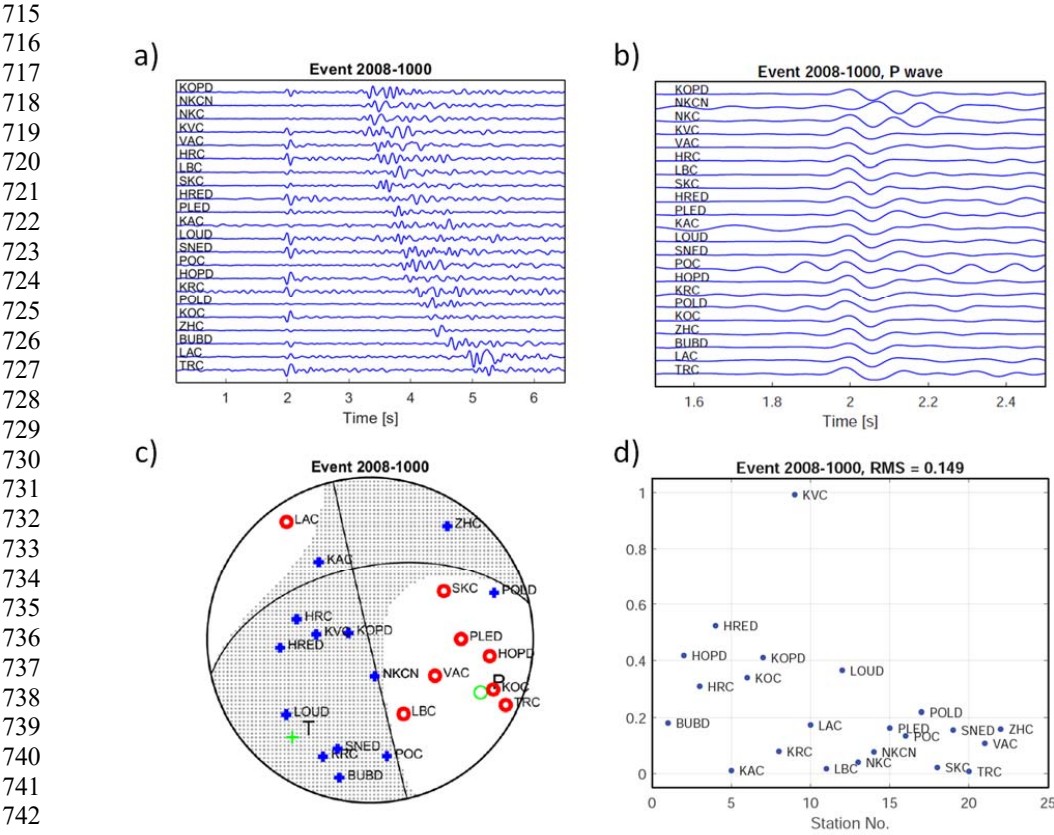

**Figure 12.** Example of plots provided for each earthquake in the dataset. (a) Vertical components of complete waveforms recorded at the WEBNET stations and aligned according to the arrival time of the P wave. Stations are sorted according to their epicentral distance. (b) Vertical components of the P waves aligned according to their arrival time and with a polarity switched according to the polarity of the common wavelet. (c) The focal mechanism with positions of the stations on the focal sphere (negative polarities – red circles, positive polarities – blue plus signs). (d) Root-mean-squares (RMS) of the differences between the theoretical and observed amplitudes of the P waves.

## 9 Discussion and conclusions

We publish a unique catalogue of moment tensors of microearthquakes that occurred in the West Bohemia in the period from 2008 to 2018. The catalogue is exceptional in several aspects: (1) it represents an extraordinary extensive dataset of more than 5.000 MTs, (2) it covers a long period of seismicity in the studied area, during which several prominent earthquake swarms took place, (3) the foci locations and retrieved MTs are of a very high accuracy. In addition, the three-component velocigrams recorded at the WEBNET stations together with



the velocity model in the region and the technical specification of stations are provided. This predetermines
the dataset to be utilized by a large community of researchers for various seismological purposes.

The great potential of the dataset or its subsets has so far been proved in studies of origins of the swarm activity
in this area (Horálek and Fischer, 2008; Fischer et al., 2010; Fischer et al., 2014), migration of seismicity in
time due to fluid flow and/or stress redistribution in the focal zone (Hainzl et al, 2012, 2016; Vavryčuk and
Hrubcová, 2017), changes of the $v_P/v_S$ ratio in the focal zone (Dahm & Fischer, 2014; Bachura & Fischer,
2016), identification of fault segments and their mutual interaction (Vavryčuk and Adamová, 2018; Vavryčuk
et al., 2021), the fault instability (Vavryčuk, 2011b, 2014), differences in the seismic energy release in
earthquake swarms and mainshock-aftershock sequences (Čermáková and Horálek, 2015; Vavryčuk and
Adamová, 2018), the efficiency of new moment tensor inversion algorithms such as the MT inversion based
on the PCA (Vavryčuk et al., 2017), the MT inversion using the empirical Green's functions (Vavryčuk and
Adamová, 2020). The provided records were also utilized in a study of seismic anisotropy based on the analysis
of shear-wave splitting (Vavryčuk and Boušková 2008), identification of shallow discontinuities in the Earth's
crust (Hrubcová et al., 2016), lateral variation of depth of the Moho discontinuity (Hrubcová et al., 2013,
2017), and for detailed mapping of the non-DC components of MTs and shear-tensile fracturing in the Nový
Kostel focal zone (Vavryčuk, 20011a; Vavryčuk et al., 2021).

The dataset is ideal for being utilized in many other studies in future, e.g., for studies of (1) the interaction
between the scattered background regional seismicity and the swarm seismicity focused in the Nový Kostel
zone, (2) the Coulomb stress and local stress anomalies connected to fault irregularities, (3) diffusivity of fluids
along the activated faults, or (4) time-dependent seismic risk due to the migration of seismicity in the region.
In addition, the dataset is optimum for developing and testing new MT inversions (Šílený and Vavryčuk, 2000,
2002), stress inversions, and for the spatiotemporal evolution of tectonic stress. Since most of the earthquakes
are non-shear, the dataset can contribute to studies of the non-DC components and their relation to shear-tensile
fracturing and/or seismic anisotropy in the focal zone (Vavryčuk, 1997; Vavryčuk and Boušková, 2008).

**Data availability**
The waveforms are available at https://doi.org/10.17632/4swk36hbvz.1 (Vavrycuk, 2021). For the review
purpose, the MT catalogue and other data are available at the following temporary link:
https://drive.google.com/drive/folders/1HyFJO6aIwN5SctwsYp-GIhERspeVVz03?usp=sharing. After the
acceptance of the paper, the temporary link will be substituted by a permanent doi number accessed under a
non-restrictive license CC BY.


**Acknowledgements**
The study was supported by the Grant Agency of the Czech Republic, Grant No. 19-06422S.



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
