# Peer review of "Moment tensor catalogue of earthquakes in West Bohemia from 2008"

_Earth System Science Data, 2021_

## Author Response (AR1)

**Comments of Reviewer 1 (in black) and our response (in red)**

**General comments**
The data presented here form a unique possibility to study a tectonically complex region of seismic swarm activity connected to crustal fluid flow on a local scale. Possible topics of further studies making use of this comprehensive set of moment tensors are manifold and will not only serve seismology but also neighbouring fields and interdisciplinary studies.

**Specific comments**
- To abstract: It would be helpful to name the magnitude range of the events included in the catalogue. Terms such as "microearthquakes" are not strictly defined. The same holds for the paragraph beginning at line 46.
  OK, corrected
- At line 76: "… microearthquakes rarely exceed a value of 4 …" Well, though, there is no clear definition of terms such as microearthquake, at least, there is a guideline once published by USGS: https://www.gns.cri.nz/Home/Learning/Science-Topics/Earthquakes/Monitoring-Earthquakes/Other-earthquake-questions/What-is-the-Richter-Magnitude-Scale (cannot find the original USGS source now) If M≤4 is the magnitude range the authors are talking about I suggest to skip the term "micro" entirely or change it to "minor earthquakes".
  OK, "micro" is skipped
- To abstract: It would also be helpful to get a brief information how these moment tensors were determined (method).
  OK, corrected
- Line 31: I would rather write "It can be separated into …" instead of "It is formed by …" because there are also other possibilities to decompose a moment tensor.
  OK, corrected
- Line 127: I would say "… with a minor azimuthal gap of X degree to the south." (At least, this is my impression from Fig. 1.)
  OK, corrected
- If someone want to make use of these data repository the provided station information are not sufficient. It needs the exact date when hardware was changes, not only the year. Also, information for instrument correction are missing. XML file format is good, modern standard.
  XML file format is included
- Line 287/288: I do not understand the message of this sentence.
  The sentence is reformulated
- 290/291: Correct! But a short reasoning why this is the case would be helpful for those who are not familiar with the topic.
  OK, explained.
- To the waveforms provided: They are provided in ascii format (four columns of time and three components of data for each event and station) and referenced/sorted according an event identification system provided in the file catalogue_2008-2018.dat along with basic catalogue information (date, time, location). That is clearly arranged and relatively easy to use. However, I am wondering if there are plans to integrate the data into international data centers such as EIDA within the EPOS Data Portal. That would guarantee an even wider visibility of the data.
- Thank you for this comment, we uploaded the dataset into the International Seismological Centre (ISC) Dataset Repository.

**Technical corrections**

- The red dots in Fig. 1b are hard to see. Maybe try a different colour?
  To highlight the active area with red dots, we marked it by a rectangle. The rectangle corresponds to the area shown in fig. 1a.
- Line 125: " … is operating …"
  corrected
- Line 148: "The most intense periods of seismicity are …"
  corrected
- Line 297: "Into a set of the so called …"
  corrected
- Line 468: I think the "range" is mixed up here. Text states it goes from 25% to 25%.
  corrected

Citation: https://doi.org/10.5194/essd-2021-349-RC1

We thank very much for the constructive review on our manuscript.

**Comments of Reviewer 2 (in black) and our response (in red)**

Manuscript essd-2021-349 presents development of seismic moment tensor catalog for Western Bohemia region. The Authors provide the vast catalog of moment tensors calculated between 2008 and 2018, as well as associate this catalog with pre-processes waveform data enabling the scientific community to perform a follow-up studies. I genuinely welcome such approach very much and appreciate

Regarding the core scientific part, i.e. the description of development of MT catalog, I have no comments. The Authors provide much more that I would expect regarding the moment tensor processing, and they also provide some more in-depth information on the peculiarities of the MT catalog. The core MT catalog is described sufficiently and it's easy to get deeper into the applied procedures by digging into the referenced studies.

(1) The organization of data is slightly confusing or alternatively I don't understand fully how Authors want to share MT catalog. There is already associated data publication [1] containing seemingly full waveform data, locations, basic station information, and velocity model. This data publication is complete and seemingly consistent internally.

The submitted manuscript provides additional supplementary information which contains already some overlapping data, i.e. velocity model and station information. There is also new information (figures and moment tensor catalogue) which both constitute the core of the supplementary information, and these are broadly described in the publication. However, I have a problem with remaining folders, i.e. waveforms and figures. These folders contain limited data (not for all MTs in the catalog), and these are certainly duplicated of what is in [1] already. I don't understand clearly (or maybe I missed something) why these limited set of waveforms is provided in the supplemnet. It makes to me more sense to provide only MT catalog and refer to [1] or even incorporate MT catalog in [1] as an update.

- The folders with waveforms and figures contained just a limited number of data for the purpose of the review process. Of course, all data and figures are provided in the final form of the paper.

- Since the revised version of the dataset adopts different names of folders for waveforms of individual events (see our response to your point 3), we finally decided to assign a new doi number to the whole dataset (the catalogue + waveforms). In fact, two doi numbers are now provided: one doi to the dataset saved in the Mendeley Data and another doi to the dataset saved in the ISC Seismological Dataset Repository.

2) The data is provided mostly as text files with the waveforms already converted to actual ground motion. It could be argued whether this is the right way to do, but I personally welcomed that the authors aimed to simplify data accessibility by removing sensor response. An alternative would be to provide data in some recognized format (mseed?) and provide sensor responses *.resp files, but as I said, I am not insisting on that. However, I would clearly expect there is some more data provided detailing this step of data processing. More specifically, how sensor response was removed, what filters have been applied etc. to obtain the signals that finally finished in waveform files. The sensor information is not sufficiently well described. The publication contains some basic sensor information, but the pre-processing of data is not described well.

We provide more detailed information about sensor response and filtering in the revised manuscript. We included the XML and dataless files containing information on stations.

3) The organization of various data types is fairly clean (i.e. clear directory structures, keeping some key information in file names and folders that allows to link various data altogether, and so on). However, the key moment tensor catalog DOES NOT contain any unique event identifier! This is actually quite surprising, because it forces the future data user to get the date of event from MT catalog, go to main catalog from data publication [1] (/locations), take the identifier from there (e.g. 'A1000') by comparing the dates and times, remember year on the way, and then, having all this information, try to build up path to the waveform data stored in [1] in another folder. This could be easily omitted by providing even the e.g. 'A1000' identifier in MT table, or actually, for example the relative path to the folder containing waveforms.

- We agree that the event identification is not well described in the original manuscript and it might be confusing. Therefore, we changed names of event directories (e.g., A1000 to 1000).
- The event identification (the first column in the catalogue) is now defined in a clear way by two numbers: year and event number. Hence, event 1000 in 2008 has its ID in the catalogue as 2008-1000. In this way, the event ID is fully consistent with that used in figures.

We thank very much for constructive comments on our manuscript.

Refs:

[1] Vavrycuk, Vaclav (2021), "WEBNET Data 2008-2018", Mendeley Data, V1, doi: 10.17632/4swk36hbvz.1

**Citation**: https://doi.org/10.5194/essd-2021-349-RC2